# Natural Selection beyond Life? A Workshop Report

**DOI:** 10.3390/life11101051

**Published:** 2021-10-07

**Authors:** Sylvain Charlat, André Ariew, Pierrick Bourrat, María Ferreira Ruiz, Thomas Heams, Philippe Huneman, Sandeep Krishna, Michael Lachmann, Nicolas Lartillot, Louis Le Sergeant d’Hendecourt, Christophe Malaterre, Philippe Nghe, Etienne Rajon, Olivier Rivoire, Matteo Smerlak, Zorana Zeravcic

**Affiliations:** 1Laboratoire de Biométrie et Biologie Évolutive, Université de Lyon, Université Lyon 1, CNRS, UMR 5558, 43 Boulevard du 11 Novembre 1918, 69622 Villeurbanne, France; nicolas.lartillot@univ-lyon1.fr (N.L.); Etienne.Rajon@univ-lyon1.fr (E.R.); 2Department of Philosophy, University of Missouri, 438 Strickland Hall, Columbia, MO 65211, USA; ariewa@missouri.edu; 3Department of Philosophy, Macquarie University, Balaclava Road, North Ryde, NSW 2109, Australia; pierrick.bourrat@mq.edu.au; 4Charles Perkins Centre, Department of Philosophy, The University of Sydney, Camperdown, NSW 2006, Australia; 5Department of Philosophy, University of Bielefeld, 33615 Bielefeld, Germany; mariaferreiraruiz@gmail.com; 6INRAE, Domaine de Vilvert Bâtiment 211, 78352 Jouy-en-Josas, France; thomas.heams@agroparistech.fr; 7Institut d’Histoire et de Philosophie des Sciences et des Techniques, CNRS (Centre National de la Recherche Scientifique), Université Paris I Sorbonne, 13 Rue du Four, 75006 Paris, France; philippe.huneman@gmail.com; 8Simons Centre for the Study of Living Machines, National Centre for Biological Sciences, Tata Institute of Fundamental Research, Bangalore 560065, India; sandeep@ncbs.res.in; 9Santa Fe Institute, Santa Fe, NM 87501, USA; lachmann@santafe.edu; 10Centre de St-Jérôme, Laboratoire de Physique des Interactions Ioniques et Moléculaires, Aix-Marseille Université, CNRS, UMR 7345, 13013 Marseille, France; ldh@univ-amu.fr; 11Centre de Recherche Interuniversitaire sur la Science et la Technologie (CIRST), Département de Philosophie, Université du Québec à Montréal (UQAM), 455 Boulevard René-Lévesque Est, Montréal, QC H3C 3P8, Canada; malaterre.christophe@uqam.ca; 12Laboratoire Biophysique et Evolution, CNRS UMR Chimie Biologie Innovation 8231, ESPCI Paris, Université PSL, 10 Rue Vauquelin, 75005 Paris, France; philippe.nghe@espci.fr; 13Center for Interdisciplinary Research in Biology (CIRB), Collège de France, CNRS, INSERM, Université PSL, 75005 Paris, France; olivier.rivoire@college-de-france.fr; 14Max Planck Institute for Mathematics in the Sciences, Inselstrasse 22, 04103 Leipzig, Germany; msmerlak@perimeterinstitute.ca; 15Gulliver Lab, CNRS UMR 7083, ESPCI Paris, PSL University, 75005 Paris, France; zorana.zeravcic@espci.fr

**Keywords:** natural selection, individuality, levels of selection, evolutionary biology, physics, philosophy of biology, exobiology, origins of life

## Abstract

Natural selection is commonly seen not just as an explanation for adaptive evolution, but as the inevitable consequence of “heritable variation in fitness among individuals”. Although it remains embedded in biological concepts, such a formalisation makes it tempting to explore whether this precondition may be met not only in life as we know it, but also in other physical systems. This would imply that these systems are subject to natural selection and may perhaps be investigated in a biological framework, where properties are typically examined in light of their putative functions. Here we relate the major questions that were debated during a three-day workshop devoted to discussing whether natural selection may take place in non-living physical systems. We start this report with a brief overview of research fields dealing with “life-like” or “proto-biotic” systems, where mimicking evolution by natural selection in test tubes stands as a major objective. We contend the challenge may be as much conceptual as technical. Taking the problem from a physical angle, we then discuss the framework of dissipative structures. Although life is viewed in this context as a particular case within a larger ensemble of physical phenomena, this approach does not provide general principles from which natural selection can be derived. Turning back to evolutionary biology, we ask to what extent the most general formulations of the necessary conditions or signatures of natural selection may be applicable beyond biology. In our view, such a cross-disciplinary jump is impeded by reliance on individuality as a central yet implicit and loosely defined concept. Overall, these discussions thus lead us to conjecture that understanding, in physico-chemical terms, how individuality emerges and how it can be recognised, will be essential in the search for instances of evolution by natural selection outside of living systems.

## 1. Introduction: Why Investigate “Natural Selection beyond Life”?

The principle of natural selection occupies a central role in biology: explaining why living organisms harbour properties apparently fitted to particular functions, and thus denoted as “adaptive”. In doing so, it provides a non-finalistic justification for “functional thinking” [1,2]; a typically biological mode of inquiry where structures, or more generally features, are investigated in light of their observed or putative effects, in interrelations with others, with which they form a functioning “whole”, the organism. Within the standard evolutionary framework, the process of natural selection is commonly conceived as the inevitable consequence of necessary and sufficient preconditions, namely “heritable variation in fitness related traits” [3] (provided it is not overwhelmed by random events). Such a formulation naturally leads one to wonder whether non biological systems may also fulfil these conditions. In turn, such an interrogation constitutes an occasion to revisit whether evolution by natural selection necessarily produces features that can be qualified as functions, that is, whether functional thinking becomes a relevant mode of inquiry whenever the conditions for natural selection are fulfilled.

Life-derived objects not typically seen as “living”, such as words, ideas or computer programs, are nevertheless endowed with some kind of self-replicating ability, and thus stand as obvious candidates for “evolution by natural selection” outside of biology. Accordingly, the principle of natural selection made its way into linguistics, cultural evolution and computer sciences [4,5]. The potential relevance of natural selection to physical sciences is a priori less obvious: since living entities are part of the physical world, concepts from the physical sciences commonly flow into biology, but the reverse flow would be unusual. Perhaps unusual but not logically impossible insofar as concepts may flow between scientific disciplines without respecting the natural hierarchy of their objects. If the a priori objection that natural selection cannot be relevant to physical sciences is thus ruled out, the question of whether it is remains largely unexplored.

This issue was at the centre stage of a workshop held in November 2019, that gathered a group of evolutionary biologists, chemists, physicists, and philosophers of science. Here we relate the major questions that were debated in this context. The article is structured as follows. We first briefly describe the research objects of connected fields, from synthetic biology to the origins of life, that we take as starting points for considering natural selection at the edge of biology. Experimentally mimicking evolution by natural selection in these fields stands as a key objective, that seems hindered not only by technical difficulties, but also by the conceptual challenge of tracking this process, identifying its conditions, and expected outcomes. We then discuss whether natural selection can be situated in the framework of far-from-equilibrium physics, a field that is explicitly aimed at encompassing living systems. Our tentative conclusion is that it cannot, perhaps because natural selection, at least as currently formalised, is at odds with the epistemic status and usage of physical principles. Next, we conversely examine whether some physical systems may be situated in the framework of evolutionary biology and argue that this will require further formalisation of the natural selection principle, to make it portable across disciplines. In particular, we highlight that natural selection requires “individuals” (i.e., well-identified units) as a precondition although, paradoxically, biological individuals are also considered as outcomes of this process. We end this report with the conjecture that solving this paradox may be a necessary first step toward the search for natural selection beyond life. This implies understanding how individuality may emerge and perhaps be reinforced in the course of evolution.

## 2. Natural Selection in Protobiotic Systems?

Although not always stated in such terms, the very possibility of applying the principle of natural selection to physico-chemical systems is a common theme of several research areas lying on the fringe of physical and life sciences: synthetic biology, exobiology, and the origins of life. While pursuing distinct objectives, these fields share a common vast technical and conceptual challenge: bridging the gap between physico-chemical and biological systems, explaining the transition from inanimate to living matter. Darwinian evolution being recognised as an important component of such a transition, many experimental and theoretical systems have been designed with the objective of mimicking this process. Such setups originally implemented some form of experimenter-assisted natural selection in the laboratory [6]. This approach led to a now well-established process of “in vitro” or “directed” evolution [7,8]. Yet such settings take for granted recurrent human interventions to handle the core feature of replication, while our focus here is on systems that replicate autonomously and are left to evolve by themselves.

Many potential candidates have been designed, falling into partially overlapping broad groups. Some are directly inspired by the template-based replication of nucleic acids seen in extant organisms. Such systems typically follow up from the discovery of naturally occurring ribozymes [9,10] and fit in a model where RNA occupies a central role in the emergence of life. The continued search for an RNA-replicase has uncovered increasingly powerful ribozymes capable of ligating up to a hundred nucleotides in emulsions, though still short of exhibiting complete self-replication capability [11,12], for a review, see [13]. Other systems take the form of autocatalytic networks and are thus more centred on metabolism as a central feature. Some are based on peptides [14,15] but others involve RNA and remain tightly connected with the RNA-world hypothesis [16,17,18]. Yet another group of protobiotic replicating systems involves more physical or self-organizing entities such as vesicles, crystals, colloids, or nanotubes [19,20,21]. Objects from these different categories may also be merged to combine their respective advantages, e.g., [13].

Many of these systems arguably display some form of self-replication. Some variations among replicating entities may also exist but only a limited fraction of them is heritable. As a result, it remains currently unclear whether such systems should be considered as evolvable by natural selection. This may reflect the technical difficulty of designing systems that will effectively display a larger diversity of heritable states. Strikingly, the challenge is also conceptual, to the point that it appears difficult to even imagine theoretical systems that would radically differ from the biological paradigm of self-replicating nucleic acids and could yet be considered as evolvable by natural selection [22,23,24]. Addressing this challenge may require shifting away from a categorical to a continuous perspective, as previously advocated in the context of the origin of “lifeness” [25,26,27,28,29]: asking, not if these systems are evolvable through natural selection, but to what extent they are, on a quantitative scale that remains to be formalised in cross-disciplinary terms. In the next section, we discuss whether the physical approach to far-from-equilibrium systems may take us closer to that objective.

## 3. Natural Selection in the Context of Physical Phenomena

Assessing the possibility that natural selection takes place in non-living physical systems first implies positioning living systems, where natural selection is recognized to happen, in the range of physical phenomena. This question has been predominantly examined from a thermodynamic perspective, starting in the 1940s with Schrödinger’s influential book [30]. We discuss more specifically the approach developed in the 1970s by the Brussels school of thermodynamics [31,32,33], which continues to this date to inspire new works [34,35]. Within this approach, as explained in the following, living systems are viewed as belonging to a larger class of open and far-from-equilibrium systems called dissipative structures, and one seeks a general framework in the form of a variational principle (as defined below), of which natural selection could be a particular case.

The term “dissipative structures” designates steady states that display spatial and/or temporal patterns (e.g., inhomogeneous distribution of chemical species, or sustained oscillations) which typically occur due to an instability from a homogeneous, patternless steady state subject to a small perturbation. Dissipative structures commonly occur in systems which are open (i.e., can exchange matter and energy with the environment), nonlinear (in terms of the governing dynamics equations) and far from equilibrium. A canonical example arises from the hydrodynamic instability known as the Rayleigh–Bénard convection [36,37]. This instability is observed in a horizontal layer of fluid heated from below. As the temperature difference between the bottom and the top is raised, a threshold is reached at which the fluid loses its spatial homogeneity and shows motions organized in patterns. This transition formally corresponds to an instability of the homogeneous state upon fluctuations (small random variations in density for instance). Depending on the difference of temperatures, the initial and boundary conditions (i.e., spatial and temporal constraints set by the environment), and the protocol followed to raise the temperature, one can observe patterns of many different forms, from simple convection cells and rolls to more complicated spatio-temporal structures. Dissipative structures are also found in reaction-diffusion systems in chemistry (Turing patterns, Belousov–Zhabotinsky oscillations), in fluid mechanics (Faraday waves) and in nonlinear optics (light beam modulations). Several biological phenomena have also been studied from the standpoint of dissipative structures, including gene regulation and biological rhythms [38]. The overarching theme in all these examples is that structures can emerge from fluctuations through instabilities when a system is maintained out of equilibrium, for instance by a gradient of temperature or an influx of chemical compounds.

From a thermodynamic perspective, living organisms constitute far-from-equilibrium open systems, but populations of organisms can also be conceived this way, and their evolution through natural selection may then be framed in terms of instability [39]. In particular, introducing a mutant in a previously stable population can be seen as provoking an instability in the population dynamics, eventually leading to a new steady state with a completely different structure, that is, a different composition of the population. Furthermore, in numerical simulations of artificial chemistries aiming at modelling the emergence of evolutionary dynamics from elementary physical rules [40,41] one observes instabilities giving rise to various dissipative structures, such as competing catalytic cycles. From this standpoint, one may therefore consider biological populations subject to natural selection as particular dissipative structures.

A major goal of the Brussels school was to identify a variational principle that generally applies to non-equilibrium steady states, and therefore to dissipative structures. In physics, variational principles take the form of mathematical functions of one or more physical quantities whose minimisation allows one to predict the final state of a system, without reference to its initial conditions or particular dynamics. For example, in equilibrium thermodynamics, this is achieved by minimizing free energy. Far from equilibrium, however, in systems such as dissipative structures, no such principle has been found [42]. In fact, it is well-established through explicit counter examples that the most likely steady state of a non-equilibrium system can depend on parameters that cannot be estimated from the immediate vicinity of stable steady states only [43]. Hence, a state-function whose optima (that is, whose minima) indicates which non-equilibrium steady states are favoured cannot be derived. In other words, dissipative structures cannot be generally predicted from summary descriptors of the steady states. A general framework exists for rationalizing a variety of patterns observed in non-equilibrium systems, but it is limited to local stability analyses [42]. In short, one can recognize that a particular steady state is unstable but not generally predict which final state it will reach.

From this standpoint, natural selection is not different. In particular cases, a fitness function can be defined in which the maximum describes the steady state of population dynamics. Many other examples exist, however, where such functions cannot be defined, for example in cases where optimal trait values depend on the composition of the populations, e.g., in systems following dynamics analogous to those of the rock-paper-scissors game [44]. This is unsurprising from the standpoint of general dynamical systems, where no steady state is even generally guaranteed. What non-equilibrium thermodynamics teaches us is that even describing biological systems in a physical framework, as subclasses of dynamical systems (e.g., by invoking physical constraints), would not be sufficient to solve this problem, that is, to understand them using a general variational principle.

From the standpoint of its epistemic status and usage, the principle of natural selection is, however, markedly different from variational principles in physics: unlike variational principles in physics, the principle of natural selection is used even in the absence of a precise quantitative description or a well-defined optimum. Trying to subsume natural selection into a variational principle, or reciprocally, may therefore be inappropriate. One may nevertheless wonder if the principle of natural selection, which is so fruitful in biology despite not constituting a valid variational principle from a physical standpoint, can provide comparable insights into physical but non-biological systems. In the following section, we discuss whether the conditions and signatures of natural selection, as depicted in the framework of evolutionary biology, can provide hints on what such systems could be.

## 4. Natural Selection as a Framework

We take here two complementary approaches to try and characterise non-living physical systems that may be subject to natural selection. One is to look for the causes of natural selection, its necessary and sufficient conditions. The other is to look for its consequences, the patterns it generates. The former may thus be labelled as “causal” or “mechanistic”, while the latter is more correlative or phenomenological, and will only provide strong evidence for natural selection if it relies on specific and unambiguous signatures.

### 4.1. The Conditions of Natural Selection

Following Lewontin [3] and many subsequent authors, natural selection is often presented as necessarily resulting from heritable differences in fitness-related traits among individuals. This Darwinian recipe provides a starting point for the causal approach, although, as abstract and general as it may sound to most biologists, it remains very much dependent upon biological concepts. Making it portable across disciplines requires more formal definitions of its components: “heritable differences”, “fitness-related traits” and “individuals”.

“Heritable differences”, first, refer to parent-offspring resemblance, and more specifically to the fraction of differences seen among individuals that stem from differences among their parents. The concept of inheritance is tightly connected to that of reproduction, although they are not strictly equivalent. Specifically, reproduction may take place without heritable differences if all individuals are identical, but heritable differences, insofar as they refer to parents and offspring, imply reproduction. It has been argued that reproduction should not be seen as a necessary ingredient of natural selection [45,46,47,48] because mere differences in “survival” among different kinds of entities, if they are persistent enough, suffice to induce deterministic changes in their relative abundances over time. Under this broadened view, natural selection is the biological name for the sorting process taking place, with or without reproduction, in any dynamical systems composed of entities differing in their inherent stability or emergence rates. As discussed in Box 1 using a toy mathematical model (and illustrated in Figure 1), reproduction nevertheless introduces radical changes in the efficiency of this sorting process, opening the possibility of otherwise unlikely cumulative changes. It may thus be recognised that reproduction is a necessary component, if not of natural selection in its most general sense, at least of “cumulative changes through natural selection”, which we take as an equivalent to “evolution by natural selection”, and to which we happily restrict our focus. The biological concept of reproduction is equivalent to that of autocatalysis in chemistry [49,50] and can also be related to exponential growth. The latter is essentially a mathematical concept but captures the dynamics of processes involving self-amplification. Yet we note that many physical systems, such as nuclear chain reactions, display exponential dynamics without being reducible to self-amplification of particular entities. In that sense, reproduction and auto-catalysis constitute, not a general equivalent of exponential growth, but rather a particular case.

If we turn to the notion of “fitness-related traits”, perhaps not encouraging is the acknowledgment that even within biology, reaching a consensual definition of this expression is probably impossible, for fitness is a notoriously slippery term [51,52,53]. Yet most biologists would probably agree that it relates to survival and/or reproduction, that condition one’s own persistence and the number of offspring one may produce. Taking only the “survival” component, the fitness of an object may be regarded as equivalent to its “stability”, a concept that seems readily applicable to any physical entity. Including the “reproduction” component breaks this equivalence. Fitness may then be related to an extended notion of stability that would apply to dynamic structures maintained although they are made of unstable components, that is, to steady states. Yet, proposals to define fitness in physical terms along those lines, including for instance dynamic kinetic stability [34] are subject to the limitations reviewed in Section 3: they cannot provide a general criterion for specifying the steady state that a system will adopt unless their scope of application is precisely and rigorously circumscribed.

The third component of the above-defined Darwinian recipe, individuality, is probably the most central although it is generally kept implicit and thus goes unnoticed: heritable differences and fitness related traits just cannot be conceived without referring to individuals. Applying the recipe approach to non-biological systems thus requires defining individuality without referring to biological concepts. Clarifying the meaning of this term within biology would probably be a good start, but this task in itself is acknowledged as very challenging [54]. Although central to virtually any biological reasoning, “individuals” are perhaps sufficiently evident in many contexts to let biologists live well without defining the underlying concept. Yet many problematic situations can also be found within biology. Vegetative propagation through cutting, or more generally clonal reproduction, is a situation where many individuals may be seen as a single individual of a higher level [45,51,55]. Obligate symbiotic associations pose similar difficulties. These border-line cases have prompted the development of more rigorous approaches to individuality [56,57,58]. In particular, the research fields of “major evolutionary transitions” and “levels of selection” provide us with useful conceptual tools to apprehend this problem in an evolutionary perspective, by describing how new levels of individuality may emerge through increased cooperation and decreased conflict among lower-level entities, turning higher levels into more effective targets of natural selection (e.g., see [56,59,60,61,62]). In this framework, individuals are thus depicted as the product of an evolutionary process, which comes as a paradox if individuals are also recognised as an essential ingredient of evolution by natural selection.

### 4.2. The Signatures of Natural Selection

When it comes to describing the outcomes of natural selection, one conceptual tool is frequently invoked: the Price equation [63]. This equation formalizes in statistical terms the notion that the change in the mean value of a biological trait across two time points (typically, between two generations) can be partitioned into an “individual-level term” (how much individuals themselves or their offspring have changed) and a “population-level term” (how much the relative abundances of the different classes of individuals have changed). Technically, both terms are defined at the population level, but the first refers to individual changes, hence the terminology used here. The population term captures the outcome of natural selection as a covariance between trait value and abundance, although this covariance may also be inflated by pure chance [64]. The Price equation is most renowned for its abstractness (thus its generality) and its robustness to particular assumptions, making it a versatile tool adapted to a wide diversity of evolutionary questions, from epidemiology to non-genetic inheritance and social or cultural evolution [65,66,67,68]. Yet, also because of its abstractness and generality, this equation remains purely descriptive and may be judged of limited utility unless it is used in a biologically well-delimited context [69].

While initially derived in reference to genetics and evolutionary biology, the Price equation was also explicitly perceived by its own author as a possible first step toward “a general selection theory” that may be used beyond its original field [70]. In principle, it may indeed apply to any dynamical system where sets of individual entities can be mapped over different time points, for example through parent-offspring relationships, or simply through conserved “identities” in non-reproducing systems. It then formalizes the notion that the average change over time in any property can be partitioned into some individual level changes and some higher-level changes, where the latter is a covariance between the property value and its relative abundance. If, under this framework, the observation of a systematic (that is, non-random) non-zero covariance, for a given property, is to be taken as a signature of natural selection on this property, one may conclude that natural selection is just everywhere, or more specifically, in any system where some form of stability is deterministically associated with some property. In that sense, using the Price equation to detect natural selection appears as an excessively permissive approach, that fails at limiting our focus on what we defined above as evolution by natural selection, which implies reproduction.

Another difficulty lies in finding the appropriate level of description. If the individual- versus population-level partitioning is always possible, the equation in itself does not help to define those levels, because of the so-called problem of “cross-level by-products” [56,71]: natural selection acting at one level will also give rise to a non-zero covariance at higher organisational levels, provided an appropriate grouping scheme. Notably, this difficulty may also occur in standard applications of the Price equation, that is, in biological systems, if different levels may be perceived as “individuals”. To account for these cases, some sophistications have been added to produce a “multi-level Price equation”. Yet, even then, any arbitrary choice may reveal elevated covariance terms at higher levels than those where fitness differences effectively take place, so that the choice of the description level requires some other kind of knowledge [56,72]. This problem is obviously inflated when it comes to considering non biological candidates for evolution by natural selection, where intuition is of no help to circumscribe individuals and populations.

In principle, multiplying the features to be measured and scanning various grains of description may provide a means to circumvent this problem: if individuals correspond to levels of organisation where selection is indeed effective, they should be identifiable as grains of description where most of the observed change can be attributed to the covariance term, and simultaneously so for a diversity of features (that is, in biological terms, for a diversity of “traits”). To our knowledge, such an approach has not been developed yet (though see discussions in [73,74]), perhaps because it implies that the features to be measured must be defined and measurable at any granularity, a constraint that does not readily fit with the intuition that individuals should display “emergent properties” that is, features that precisely cannot be defined at all levels of description. The information theory may offer a possible way out of this difficulty, by providing a means to “measure” individuality in non-biological terms [75], but the very feasibility of this approach is also questioned [73]. More generally, we note a connection between this problem and the field of coarse graining, where one aims to determine the optimal levels of description to characterise a system [76]. If natural selection is taking place, this objective may be much akin to that of defining individuals and populations.

## 5. Conclusions: Individuality beyond Life?

Building on the conception that natural selection should follow from some necessary and sufficient conditions, our discussions aimed at exploring the possibility that this process could take place beyond life as we know it, that is, in other physical systems where these conditions would be met. A short survey of protobiotic systems revealed how much evolvability through natural selection is perceived in this research field as an important yet unattained objective, perhaps because of an excessively categorical scheme, as opposed to continuous, which we take as evidence that the challenge is not only technical, but also conceptual. Considering the problem from a physical perspective, we discussed the possibility of placing natural selection within the context of dissipative structures. While this framework makes it possible to situate life among other physical phenomena, it has not produced general principles, of which natural selection might have been a particular case, reflecting that current physics does not include a readily usable equivalent to natural selection. Turning to evolutionary biology, we asked whether the conditions for natural selection, or its signatures, were defined with sufficient formalism to be identified outside of their original context; we contend they are not. Most strikingly, the implicit but essential notion of individuality stands as a major conceptual obstacle. How can individuals be recognised without *a priori* knowledge of the appropriate level of description? How can individuals be at the same time considered as essential ingredients and outcomes of evolution by natural selection? Addressing these paradoxical questions appears as an essential prerequisite for further investigating where and how natural selection may take place.

In fact, the question of how individuality may emerge resembles one that has attracted much attention in the field of major evolutionary transitions: how and why, in the history of life, have individuals merged into higher organisational levels, such as procaryotic symbionts into large eucaryotic cells, or clonal cells into multicellular organisms [54,56,59,71]? Yet in the case of the very first emergence of individuality, low level individuals cannot be part of the initial conditions. In other words, a general theory for the emergence of individuality cannot, by definition, rely on individuals. In biology, explaining jumps in levels of individuality implies identifying conditions, such as limited spatial diffusion or relatedness, that generate interdependence among the fitness of various entities, setting the stage for the evolution of cooperative traits [77]. This mode of reasoning has recently been applied to very early stages of life evolution [78] and may also help in explaining the very origin of individuality. Yet at that stage it remains unclear how to even model these questions without assuming the existence of some kind of well-delimited self-replicators, that is, without starting from individuals of lower levels. Future work will hopefully clarify whether the biological concept of individuality, and the biological principle of natural selection, can be grounded in physico-chemical roots, to perhaps extend the breadth of their applicability.

Box 1Mere sorting versus evolution by natural selection.One way to discuss whether natural selection should be seen as equivalent to mere sorting is to assess the efficiency of a sorting process with or without reproduction. This can be performed with a simple mathematical model (equivalent to those previously used by Earnshaw-Whyte [79] and Bourrat [80]) simulating the dynamics of a system composed of two or more types of entities, only differing in their rates of decay, the equivalent of “survival” in biology. Specifically, let us consider a system of green and blue entities, produced at the same rate but differing in their respective stabilities, e.g., with 80% chance of being maintained at each time step for the green kind, and only 40% for the blue kind. We assume no transition between the two types, that is, the colour (and thus the degree of stability) of the entities does not change.Starting from an equal proportion of the two, Figure 1 shows what would then happen in two situations: one without reproduction (small points), where the lost entities are replaced with green or blue ones with equal chance, versus one with reproduction (open circles) where the lost entities are more often replaced by the most abundant type, with a probability that equals its frequency (equivalent to choosing a random entity to reproduce). Without reproduction, green entities dominate the system at equilibrium, but the blue ones still constitute a fifth of the population. In contrast, with reproduction, the blue entities are just not present at equilibrium if one assumes a finite population size (1000 in these simulations). These radically different dynamics also mean that reproduction increases the chances of cumulative changes, where each new step is facilitated by the very high abundance of the fittest type. In sum, it remains theoretically possible to envisage a process akin to natural selection without reproduction, but reproduction so radically changes the dynamics that it introduces the possibility of otherwise very improbable cumulative changes. In that sense, reproduction may be considered an essential ingredient, perhaps not of natural selection in its broadest sense, but of evolution by natural selection, implying cumulative changes.

## Figures and Tables

**Figure 1 life-11-01051-f001:**
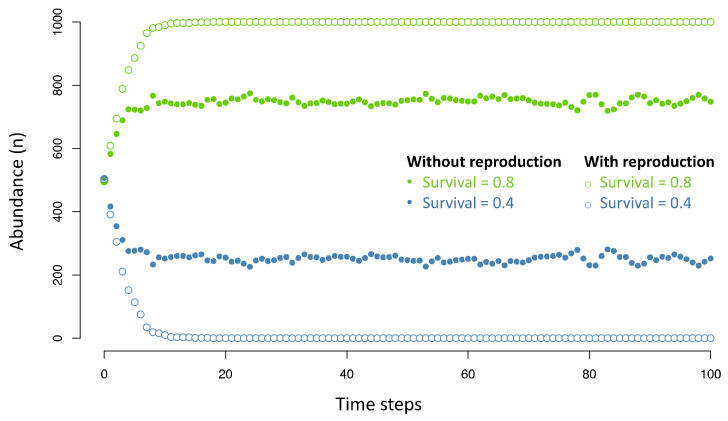
Efficiency of a sorting process with or without reproduction. Without reproduction (full bullet points), the fittest entities (in green) dominate the system at equilibrium, but the blue ones still constitute a fifth of the population. In contrast, with reproduction (empty bullet points), the blue entities are just not present at equilibrium if one assumes a finite population size (1000 in these simulations).

## Data Availability

Not applicable.

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
