# Peer review of "Natural Selection beyond Life? A Workshop Report"

_life, 2021, doi:10.3390/life11101051_

Round 1
Reviewer 1 Report
The authors reported the major questions that were debated during a three-day workshop devoted to discussing whether natural selection may take place in non-living physical systems. The topic is very interesting however many sentences result less clear or too general. In my opinion, this work is too theoretical with little evidence and many conjectures. I think that this article dint’ fit with the aims of this Journal.
Additional comments
Line 107 inanimate or non-living?
Line 110 “some form of experimenter assisted natural selection in the laboratory” Please could yo clarify these concepts it is very general sentence.
Line 115 “Many potential candidates…” please coul you provide other detail they are chemical compounds, enzymes, macromolecules or what else?
Line 137 “asking, not if these systems are evolvable through natural selection, but to what extent they are, on a quantitative scale that remains to be formalized in cross-disciplinary terms.” It is a hard sentence, you declare that these systems follow the “natural selection” please could you better support your sentence?
Reviewer 2 Report
This is a good review, and I do not have comments for improvement.
Author Response
Thanks for this positive comment.
Reviewer 3 Report
The proposed paper is a synthesis of multidisciplinary reflections (biology, chemistry, physics and philosophy) conducted during a workshop held in 2019. It proposes to answer the question: Natural section beyond life?
The article is clearly constructed and the contributions of each discipline are presented in an interesting way.
The first question posed is that of natural selection in prebiotic chemistry and it is judiciously suggested that in this case a continuum rather than a categorical approach might be useful.
The paper then turns to natural selection in the case of physical phenomena. It concludes that the principle of natural selection is radically different from the variational principles of physics, but the authors then extend the reflection by asking whether: "the principle of natural selection, which is so fruitful in biology despite not constituting a valid variational principle from a physical standpoint, could provide comparable insights into physical but non-biological systems"
In a second step, the article studies the characteristics of physical systems according to a double approach.The first is causal (or mechanistic) and studies the conditions of natural selection: "heritable differences", "fitness-related traits" and "individuals". The first is causal (or mechanistic) and studies the conditions of natural selection: "heritable differences", "fitness-related traits" and "individuals". The second approach is correlative or phenomenological and is concerned with consequences.
This interesting and stimulating part shows that the interdisciplinary discussion was successful during the workshop. Indeed, this part is based on a methodology of philosophical analysis allowing to explore the problematic posed.
In conclusion, the authors insist on the importance of the emergence of individuality and propose to ask the question "Individuatlity beyond Life?
To conclude: The authors of this article have been able to implement a convincing interdisciplinary reflection and to propose a synthesis and a stimulating and fruitful questioning. The article is an assessment of a workshop, but it is also programmatic by opening on a new question "Individuatlity beyond Life?" and it allows to understand why this one is essential. It is to be hoped that the authors will extend their collaboration and propose an analysis.
As a report of a workshop the article can be accepted in present form.
Author Response
Thanks for these positive comments.
Reviewer 4 Report
Dear Authors and Editors,
Thank you for the opportunity to review this manuscript. As this is a meeting report and not a standard contribution, I’m not going to request revisions. This should be published. It needs copy editing.
That said, there is one part of the conversation reported here that seems off to me, so I would like to bring it to the attention of the authors, either for clarification in this report, or just for them to think about. The authors say:
“we highlight that natural selection requires “individuals” (i.e.well-identified units) as a precondition although, paradoxically, biological individuals are also considered as outcomes of this process.” (95-96)
And further:
“The third component of the above-defined Darwinian recipe, individuality, is probably the most central although it is generally kept implicit and thus goes unnoticed” (278).
The authors then suggest that “intuition” is what biologists employ to locate individuals.
As Hull, Godfrey-Smith, Guay, Pradeu, and many others have pointed out, intuition is not how we carve the biological milieux into individuals. We instead employ biological theory to locate the individuals - this is called “theoretical individuation” (Gauy and Pradeu 2016), or individuation in scientific practice (Beno, Chen, and Fagan 2018). From the former perspective, when looking for evolutionary individuals, we must detect what is evolving (the “unit of evolution”) and what is being selected (the “unit of selection”). While some of the authors of this meeting report have focused beyond the standard Darwinian cases, paradigm Darwinian individuals (units of selection) are reproducers that generate parent-offspring lineages. To locate the evolutionary individuals then, we must first find something that is making parent-offspring lineages. Once we find such an individual, we can measure its fitness by noting how many of these offspring lineages are generated before it is eliminated, so multiplication selection and elimination selection. A group of such units of selection propagating together as one evolving thing, especially in case of a group of sexually-recombining tokogenetic lineages, is a Darwinian population lineage or “unit of evolution”. Individuality follows, then, from the role played in evolutionary process, not intuition. The opening Chapter of Guay and Pradeu (2016) “Individuals across the sciences” is probably the best resource for theoretical individuation. “Intuition”, or as Guay and Pradeu say “phenomenal individuation”, is not the way to locate biological individuals.
So, we have to look for reproducers if we want to find evolutionary individuals, and as the authors point out, this can lead to paradox. How do we determine whether something is reproducing if we do not yet first identify it as an individual? This is where scientific practice comes in. Jim Griesemer (2018) has addressed this question – we must first choose something to track (this choice may be arbitrary or based on intuition), and then we watch to see if it reproduces. If it does reproduce, then we can assume (defeasibly of course) that other things like it are similarly reproducers. Griesemer notes that biological reproduction always entails material overlap, so material overlap is something that we can track, with radioactive markers for example, as an indicator of reproduction and hence Darwinian individuality.
Now, I don’t think material overlap is necessary for reproduction in non-biological evolving lineages. A song, for example, can be said to reproduce when it is sung by different bands whose members do not overlap, so material overlap might not be a suitable tracking method in all evolving systems – we might instead track series of phonemes or chord progression – but the point is, individuality is not a matter of intuition; it’s a matter of using scientific theories and scientific practices to carve nature at its joints, noting of course that there are as many cross-cutting joints as there are accurate theories and successful scientific practices.
References
Bueno, O., Chen, R. L., & Fagan, M. B. (Eds.). (2018). Individuation, process, and scientific practices. Oxford University Press.
Guay, A., & Pradeu, T. (Eds.). (2016). Individuals across the sciences. Oxford University Press, USA.
Griesemer, J. (2018). Individuation of developmental systems. Individuation, process, and scientific practices, 137.
Author Response
Thanks for sharing these thoughts and drawing our attention to these resources. We agree that our previous text could give the wrong impression that intuition is the only tool at hand. We have thus integrated the following sentence (Line 301): “These border-line cases have prompted the development of more rigorous approaches to individuality [56–58]”, citing here the opening Chapter of Guay and Pradeu (2016), that indeed offers a good overview of these approaches. The “levels of selection” framework is now presented as one such approach.
Reviewer 5 Report
The article analyses the possibility to apply the concept of selection outside the field of biology. It is interesting, well written and well structured, and it offers an exhaustive list of references. Being a proceeding of a workshop, I suppose it just reflects what was discussed in it.
My main comment is that the discussion does not explicitly differentiate several uses of the concept of selection:
- selection in a non-reproducing population
- evolution by selection in a reproducing population with fixed ‘phenotypes’ (meaning offspring are identical to parents) with (a) or without (b) resource limitations
- cumulative evolution of evolving individuals (meaning there are fitness changes between generations) in a reproducing population with resource limitations, resulting in new traits performing adaptive functions.
and does not separate at least between two concepts of fitness:
- fitness as measure of surviving ability
- fitness as measure of reproducing ability
Such differences might be purely theoretical in case of biological systems, but they are key if one wants to apply the idea of selection to non-living systems, where the meaning of 'reproduction', 'adaptation' or 'function' are not evident, resources are possibly unlimited, and fitness as ‘stability’ is probably more relevant than fitness as number of offspring. Starting a quest for selection outside of biology without defining what these fundamental concepts mean in these new fields seems daring to me, and might generate confusion.
Box 1 provides an example of such confusion, which springs from the implicit use of different meanings of selection: “One way to discuss whether natural selection should be seen as equivalent to mere sorting is to assess the efficiency of this sorting process with or without reproduction.” (lines 399-400). Selection without reproduction is case 1 above (and it is not clear what the ‘time step’ axis represents in this case: the repetition of the sorting upon the same initial population?). Selection with reproduction is case 2a, given that an additional hypothesis about limitation of resources is also added (assumption of finite population). This additional hypothesis is what really differentiates this case form the previous one: without resource limitations, the less fit phenotype would never disappear. So, if ‘efficiency’ is defined as eliminating the less fit phenotype, it is resource limitation and not reproduction that guarantees it. On the other hand, Darwinian evolution in living systems is really about case 3. So, it is not clear which concept of evolution and selection the authors are trying to identify in the different types of non-living systems they analyse.
In conclusion, although the article could be published as it is, I believe it would become clearer if a few paragraphs were dedicated to such clarifications - but I leave this decisions to the authors.
Author Response
We agree that making these distinctions would make it possible to clarify how much reproduction versus resource limitation affect the chances of extinction of the less fit type (case 1 versus 2) and that the introduction of new mutation is required to reach case 3. Yet our main objective with box 1 is simply to illustrate that reproduction alone (keeping everything else equal) is sufficient to produce a radical change in the dynamics.
Round 2
Reviewer 1 Report
In my opinion, some topics have been treated in a somewhat superficial way. however, I consider the answers of the authors exhaustive and the manuscript could be published in this journal